biomechanics/physiology

bimodal, force platform, vertical jump, yank-time

**Author for correspondence:**
Sofyan B. Sahrom
e-mail: sofyan@sofyan.com.sg

# The use of yank-time signal as an alternative to identify kinematic events and define phases in human countermovement jumping

Sofyan B. Sahrom, Jodie C. Wilkie, Kazunori Nosaka and Anthony J. Blazevich

Centre for Exercise and Sports Science Research (CESSR), School of Medical and Health Sciences, Edith Cowan University, 270 Joondalup Drive, Joondalup, Western Australia 6027, Australia

(iD) SBS, 0000-0003-1717-9679

Detailed examinations of both the movement and muscle activation patterns used by animals and humans to complete complex tasks are difficult to obtain in many environments. Therefore, the ability to infer movement and muscle activation patterns after capture of a single set of easily obtained data is highly sought after. One possible solution to this problem is to capture force-time data through the use of appropriate transducers, then interrogate the signal's derivative, the yank-time signal, which amplifies, and thus highlights, temporal force-time changes. Because the countermovement vertical jump (CMJ) is a complex movement that has been well studied in humans, it provides an excellent preliminary model to test the validity of this solution. The aim of the present study was therefore to explore the use of yank-time signal, derived from vertical ground reaction force-time data, to identify and describe important kinematic (captured using three-dimensional motion analysis) and kinetic events in the CMJ, and to relate these to possible muscle activation (electromyography) events that underpin them. It was found that the yank-time signal could be used to accurately identify several key events during the CMJ that are likely to be missed or misidentified when only force-time data are inspected, including the first instances of joint flexion and centre of mass movement. Four different jump profiles (i.e. kinematic patterns) were inferred from the yank-time data, which were linked to different patterns of muscle activation. Therefore, yank-time signal interrogation provides a viable method of estimating kinematic patterns and muscle activation strategies in complex human movements.

# 1. Introduction

Detailed examinations of both the movement and muscle activation patterns used by animals (including humans) to complete complex tasks usually require the synchronous use of videographic, electromyographic and force measurement methods. They are thus often difficult to obtain in many environments. Therefore, the ability to infer movement and muscle activation patterns after capture of a single set of easily obtained data is highly sought after. Because changes in the motion of a body must relate directly to the external forces acting on that body, one theoretical opportunity is to infer movement kinematics—and hence the underlying muscle activation patterns, muscle forces and joint moments—from force-time data [1,2]. Because of the indeterminacy problem relating to redundant peripheral degrees of freedom [3] even the most superficial description of complex movements has yet to be reliably obtained from force-time data alone.

The time rate of change (instantaneous derivative) of amplitude-based (rather than frequency-based) data has been used in many fields including mechanics and acoustics to more clearly identify points of interest in a data record [4,5] (table 1). A crucial benefit of derivation during the recording of forces during animal locomotion is that small fluctuations in the force-time record can be more easily observed (figure 1), and directionality of the changes in force can be readily assessed. Since ground reaction force data can be easily obtained during self-propelled animal locomotion by use of force platforms, yank-time records computed by derivation of force-time data (i.e. the instantaneous rate of force development; RFD), can be obtained under many conditions. However, the utility of yank-time data for inference of important kinematic, kinetic and muscle activation events during complex movement tasks has yet to be determined.

A complex human movement task that has been well studied from a force-time perspective is the vertical countermovement jump (CMJ) [11,12]. Complexity of the force-time record for the CMJ results from (i) the need to achieve maximal impulse (external force production) through the optimal coordination and timing of force production from muscles in both arms and legs, which is reflected by the body's velocity at take-off and which in turn determines the jump height [13,14], and (ii) the need to decelerate the body during the downwards phase (countermovement) and subsequently accelerate it upwards to ultimately project the body into the air [12]. It therefore represents a good model in which to assess the accuracy of methods by which kinematic, kinetic and muscle activation patterns might be estimated from force-time data.

Several differences in force-time characteristics other than magnitude and timing, such as the waveform shape (e.g. unimodal, bimodal) have been observed in the CMJ. These differences have been attributed to variations in jumping strategies [6,7] and individual characteristics unique to that individual [8]. These variations result from the wide range of task adaptations that humans can adopt in response to specific constraints or as a result of interventions (e.g. different exercise training programmes of sprint versus endurance athletes) and therefore have very different movement patterns and strategies when performing the same movements, such as the CMJ, with the same objectives (i.e. maximizing vertical displacement). Existing examination methods [9,15] used to understand the unique movement patterns and strategies of a single animal/individual from force-time signal typically depend on the identification of kinematic and kinetic events to calculate numerous variables. These variables include peak force, peak eccentric (braking) force, peak power, rate of propulsive force development and modified reactive strength index [6,16]. Nonetheless, without being able to accurately determine the timing of important kinematic and kinetic events during a task such as jumping, accurate calculations of these variables are not possible. For example, in CMJs the point of onset of muscular force development (after the relaxation phase that permits the fall of the body in the countermovement), the identification of the start and end points of eccentric (downwards or countermovement) and concentric (upward or propulsive) phases, and the identification of the transition point between these two phases, are commonly mistaken. A re-examination of event definitions through the lens of the yank-time signal might reduce error and provide for more accurate calculation of variables used to describe dynamic, multi-joint movement performance.

Given the above, the first aim of the present study was to explore the use of yank, the time-derivative of force, to directly identify and describe important kinematic and kinetic events during the CMJ without the need for direct motion capture as an alternative to existing methods. Subsequently, the second aim of the paper was to relate these kinematic and kinetic events to possible muscle activation patterns that underpin them and determine whether the yank-time data can therefore be used to identify which of several broad jump strategies (techniques, as determined from the shape of force-time profiles) are used by an individual. Through these aims, we tested the hypothesis that kinematic and muscle activation patterns could be estimated from force-time data captured during a complex, multi-joint movement task.

**Table 1.** Relationships between, and calculations of, kinematic and kinetic terms. Kinematics describes the displacement or change in position. The higher-order time derivatives of displacement were summarized from the data from multiple sources [6–9]. Kinetic terms such as force and yank describe the rate of change in the momentum.

| order (in relation to time) | kinematic | | kinetic | | |
| --- | --- | --- | --- | --- | --- |
| | term | calculation | term | calculations | |
| 6th | Pop ($p$) | $\Delta c/\Delta t$ | Shake ($Sh$)[a] | $m \times p$ | $\Delta S/\Delta t$ |
| 5th | Crackle ($c$) | $\Delta s/\Delta t$ | Snatch ($S$)[a] | $m \times c$ | $\Delta T/\Delta t$ |
| 4th | Snap/Jounce ($s$) | $\Delta j/\Delta t$ | Tug ($T$)[a] | $m \times s$ | $\Delta Y/\Delta t$ |
| 3rd | Jerk ($j$) | $\Delta a/\Delta t$ | Yank ($Y$) | $m \times j$ | $\Delta F/\Delta t$ |
| 2nd | Acceleration ($a$) | $\Delta v/\Delta t$ | Force ($F$) | $m \times a$ | $\Delta P/\Delta t$ |
| 1st | Velocity ($v$) | $\Delta d/\Delta t$ | Momentum ($P$) | $m \times v$ | |

Higher-order time derivatives

Factor in (multiplied by) mass ($m$)

[a]These terms have not yet been standardized [9].

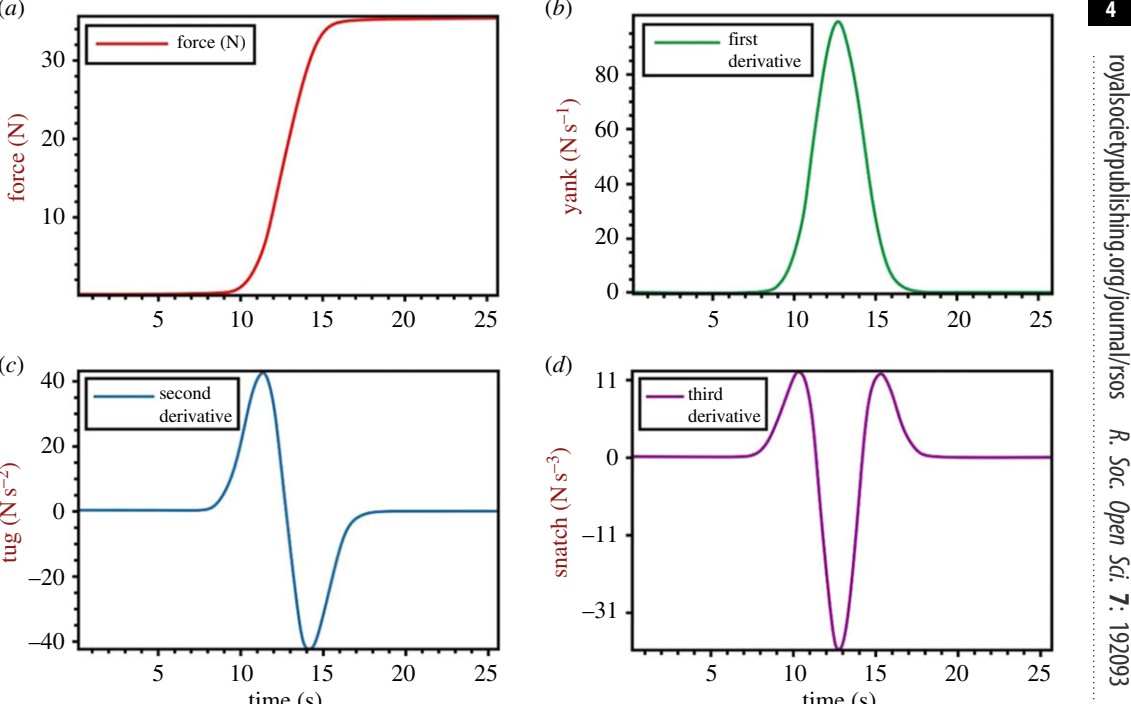

**Figure 1.** Force-time signals and subsequent derivatives. Panel (*a*) shows an original force-time signal when a load is placed on a force platform. Panels (*b–d*) show the yank-time, tug-time and snatch-time signals, which are first, second and third time derivatives of the force-time signal (panel *a*). The complexity (the number of extrema [maxima and minima] and points of crossing the *x*-axis, i.e. where x = 0) of the signal-time relation increases with derivative order. Image is taken from Sahrom, S. [10] 'Beyond jump height: Understanding the kinematics and kinetics of the countermovement jump from vertical ground reaction force data through the use of higher-order time derivatives'.

# 2. Material and methods

## 2.1. Experimental overview

Thirty-two physically active, healthy adult males (age = 25 ± 3 years, body mass = 72.0 ± 15.3 kg) who were free from lower-limb injuries volunteered for the study. Their physical activity engagement ranged from a minimum of performing deliberate exercise or sports sessions at least twice a week to international-level athletes in running-based sports. The subjects reported to the laboratory on 2 days for a familiarization (Day 1) and experimental session (Day 2), with at least 6 days separating the two. Between the sessions, the subjects completed at least two additional unsupervised jump practice sessions. Each unsupervised session commenced with a standardized warm-up (described below) before at least 15 CMJs were performed. This allowed subjects to familiarize themselves with (i) the standardized warm-up, and (ii) jumping with the hands on hips.

## 2.2. Familiarization session (Day 1)

The subjects commenced their warm-up with a 5 min jog around the laboratory at a self-selected intensity that was greater than 50% perceived maximum effort, before completing a 5 min dynamic warm-up consisting of self-selected activities (e.g. leg swings, lunges). They then performed 10 bodyweight squats (with hands on hips) and 10 standing calf-raises at a 'slow pace', before performing the same tasks with the instruction to 'move through your maximum range of motion but move as fast as possible without losing balance' with a 60 s rest between the bouts. They finally performed 10 squat repetitions pushing up into calf raise (i.e. combined movement) with a 1 s pause at the end of the calf raise phase, before performing three maximal countermovement jumps (CMJs) interspersed with 10 s of rest.

To determine the preferred stance width for jumping, the subjects performed five double-leg hops on the spot with hands on hips and eyes closed. Jump stance was determined as the distance between feet measured to the most anterior portion of the shoe on landing after the last hop. This jump stance was

maintained throughout all jump trials and types (e.g. deep or shallow). These steps were taken to try to maximize individual jump performance by allowing some individual preferences (e.g. selection of dynamic warm-up exercises) while still maintaining consistency between subjects. Stance width was commonly slightly wider than hip width and never exceeded shoulder width, based from the alignment of the heel.

CMJ familiarization included the performance of at least five CMJs as high as possible with hands on hips and to a self-selected countermovement depth before at least five jumps were performed to a greater or lesser depth. This allowed subjects the opportunity to determine for themselves whether their initial (self-selected) countermovement depth was ultimately favoured. No jump height feedback was given during familiarization; therefore, the final, self-selected countermovement depth was subjectively determined by the subjects. In the days between the familiarization and the experimental (testing) sessions, the subjects performed another two sessions similar to the first familiarization session (i.e. standardized warm-up and 15 CMJs) in their own time outside the laboratory.

## 2.3. Experimental (testing) session (Day 2)

Upon arrival, preparations were completed for three-dimensional motion analysis and electromyography (EMG) data collection, as described below (see §2.4). The subject then completed the standardized warm-up and proceeded to perform at least three CMJs with near maximal effort before taking a 3 min rest. They then commenced each trial by standing on the force platform, where they were instructed to remain stationary for approximately 3 s before being given the clearance to perform each CMJ trial. The subjects performed at least three CMJs with 30 s of rest between attempts. Up to two further attempts were allowed if either the jump height (based on the flight time) improved by more than 5% on the final jump or if the jumper believed they could jump higher. All subjects self-selected their own shoe size and wore the same generic shoe type (canvas, lace-up with flat synthetic outsole) to avoid an effect of shoe type on force production characteristics and jump height [17]. The trial with the best jump height was selected for further analysis.

## 2.4. Kinematics, ground forces and muscle activities of the lower limbs during countermovement jumps

Lower-limb kinematic and ground reaction force data during CMJs were captured synchronously using the VICON Nexus Software (Vicon NEXUS; Vicon, Oxford, UK). Kinematic data were obtained using a nine-camera Vicon three-dimensional motion analysis system (Vicon MX, Vicon, Oxford, UK) sampling at 250 Hz. Subjects wore skin-tight shorts with no shirt to allow 19, 25 mm retro-reflective skin-based markers (for three-dimensional motion analysis) to be fixed to the 7th cervical vertebrae, and on the right and left, anterior and posterior superior iliac spines, greater trochanter, lower one-third of the thigh, flexion–extension axis of the knee, lower one-third of the shank, lateral malleolus, calcaneous (heel), second metatarsal head (toe). The infrared cameras captured the position of the reflective markers, which were reconstructed into three-dimensional coordinates in the software. Raw three-dimensional coordinate data were filtered using a Woltring (MSE 10) filter [18,19], and ankle, knee and hip joint angles, angular velocities and moments as well as the body's centre of mass (CoM) displacement were subsequently calculated, as described in detail in Results. Pelvic movement was observed by creating a simplified model connecting both left and right anterior superior iliac spine, posterior superior iliac spine and greater trochanter, while CoM vertical trajectory was estimated from an average marker in the centre between the two anterior superior iliac spine and posterior superior iliac spine markers and confirmed with knee flexion. Left and right knee flexion and extension were determined from the respective thigh, knee and tibia markers, while ankle flexion was determined from the thigh, ankle and toe markers. Ground reaction force data were obtained using a 600 × 900 mm triaxial force platform (Kistler Quattro, Type9290AD, Kistler Instruments; Switzerland) sampling at 1000 Hz. The force platform was zeroed before the subject stepped on it, and data were captured from approximately 3 s prior to jumping.

Activities of eight muscles were recorded from the right side of the body during jumping (Zero-Wire, Delsys, MA, USA) using surface EMG electrodes placed in a bipolar configuration over erector spinae (ES), gluteus maximus (GM), vastus lateralis (VL), vastus medialis (VM), bicep femoris (BF), medial gastrocnemius (MG), tibialis anterior (TA) and soleus (SOL) with an inter-electrode distance of 20 mm,

in accordance with SENIAM guidelines. EMG data were acquired at an analogue-digital conversion rate of 1000 Hz through the VICON Nexus software.

# 3. Data analysis

Scientific computing tools for Python (SciPy v.1.1.0) [20] were used for all calculations and data processing (e.g. filtering, signal onset detection, residual signal analysis) while statistical analyses were performed using R (v. 3.2.5) [21], as described in detail in Results.

## 3.1. Signal processing

Residual signal analysis [22] was performed on each trial for both vertical ground reaction force ($GRF_z$) and EMG data to determine the appropriate low-pass cut-off filter for the respective signals. Both $GRF_z$ and EMG data were filtered using a fourth-order, zero-lag Butterworth filter; $GRF_z$ data were filtered using cut-off frequencies of 59–68 Hz while EMG data were filtered using band-pass limits of 10.2–13.2 Hz (high-pass limit) to 400 Hz (low-pass limit).

For the purpose of EMG onset detection ($EMG_{on}$), EMG data were subjected to a Teager–Kaiser energy operator (TKEO), which highlights spikes in a signal by increasing sharpness while maintaining the signal's temporal characteristics and has been shown to assist with onset detection with EMG signals [10,23–25]. While absolute amplitude may be slightly depressed, the point at which the EMG amplitude increases is well preserved, which is critical for $EMG_{on}$ detection. This allows a lower amplitude threshold to be used for onset detection; in the present study, $EMG_{on}$ was determined by first identifying the point at which the amplitude of the EMG signal increased and remained above 10% of the standard deviation of the baseline signal for a minimum of 50 ms and subsequently tracing backwards from that point to the origin of the EMG increase. Visual inspection (and confirmation) for each trial was also done to ensure that the automated onset detection and other EMG characteristics were correctly identified.

## 3.2. Body mass, kinematic and kinetic variables

The subject's mass was determined by averaging $GRF_z$ for 2 s during the motionless period, i.e. when the subject remained stationary, immediately prior to the start of each individual trial, and dividing by gravitational acceleration ($9.80665 \, \mathrm{m \, s^{-2}}$) [26]. Three kinematic variables and two kinetic variables were subsequently calculated from the $GRF_z$-time data (figure 2): (i) resultant vertical force ($F_z$) was computed by subtracting the subject's weight from $GRF_z$, (ii) vertical acceleration of the centre of mass ($a_{CoM}$) was computed by dividing $F_z$ by $m$, (iii) vertical velocity of the centre of mass ($v_{CoM}$) was computed by integration of the $a_{CoM}$-time data, then (iv) vertical displacement of the centre of mass ($d_{CoM}$) was subsequently computed by integration of the $v_{CoM}$-time data, and finally (v) yank-time was computed through differentiation of the $F_z$-time data using the second-order central differences method and second-order accurate difference (one-side) for the boundaries of the signal. The yank-time data were subsequently rectified to allow for definitive identification of the crossover point, i.e. where yank was equal to zero and smoothed with a Savitzky–Golay smoothing filter with a second-order polynomial [27].

## 3.3. Identification of jump phases

The $GRF_z$-time signal was divided into three phases: (i) downward phase, (ii) upward phase, and (iii) flight and landing phases (figure 2 and table 2). These terms were used in place of other commonly used terms (e.g. eccentric/concentric, braking, propulsive) as they reflect the actual movement (i.e. downwards and upwards) of the body's CoM during the CMJ (table 2 and figure 2) and minimize confusion or incorrect use of terms when describing the events occurring during the CMJ.

The first, downward, phase is commonly referred to as the countermovement or eccentric phase. It spans the time from jump initiation to the point at which the CoM has reached its lowest point. The upward phase commences when the CoM is at the lowest point and ends at the point of take-off. The final (flight) phase spans the time from the take-off to landing. These phases can be further sub-divided into nine distinct events or sub-phases (table 2). In the present study, the $GRF_z$-time signal was analysed with the aid of the computed mechanical variables as well as points such as minima

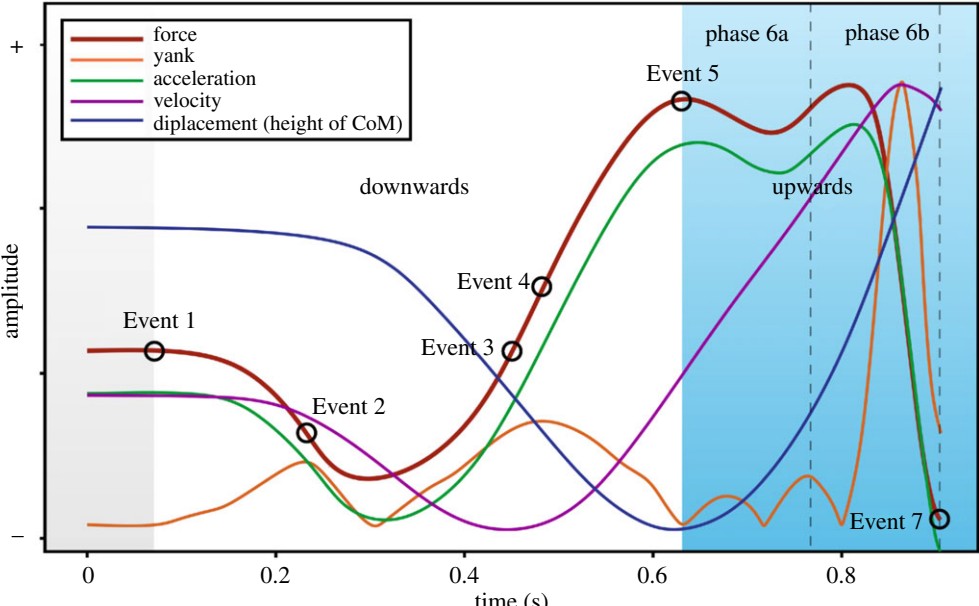

**Figure 2.** Vertical ground reaction force (GRF$_z$)-time trace, with important kinematic events and jump phases identified (flight phase not shown), and the associated displacement, velocity, acceleration and yank-time traces from one subject. Five events are shown: Event 1, start of CMJ/first instance of movement; Event 2, meaningful CoM movement (downwards); Event 3, start of braking (deceleration); Event 4, peak yank braking; Event 5, point of lowest CoM height. The upwards phase, shown as phase 6, is divided into two sub-phases: 6a, the first 50% of the upwards phase; and 6b, second 50% of the upwards phase. Further information regarding the events and sub-phases is presented in table 2.

(smallest value) and maxima (largest value, or peak) within either (i) a specific time interval or phase of the jump (local), or (ii) the entire jump duration (global) (figure 2 and table 2).

# 4. Results

The derived, rectified yank-time signal in conjunction with the original GRF$_z$-time signal were compared with data derived from three-dimensional motion analysis. It is observed that the extrema (maxima and minima) within the yank-time signal tended to align closely with events identifiable on the GRF$_z$-time signal described in table 2. These extrema and alignment with events were subsequently explored in detail, as described in the following sections.

## 4.1. Event 1: start of CMJ/downward phase

The first event (Event 1) is the start of the jump, which is a critical event, as it sets the initial force conditions for the jump phase and is commonly used in the calculation of variables such as rate of force development, time to peak force, reactive strength index (modified), etc. The start of the jump is typically defined as the first instance of movement of the body's CoM [15,28]. Because a downward movement of the body's CoM should correspond to a change in GRF$_z$, Event 1 is often also defined as (i) the point at which GRF$_z$ first starts to fall by a given multiple of the standard deviation of the baseline signal [29,30], (ii) a decrease in GRF$_z$ as a percentage of bodyweight [15], or (iii) an absolute change in GRF$_z$, e.g. when GRF$_z$ decreases by 10 N [31]. These methods equate the first decrease in GRF$_z$ with the negative (downward) acceleration of the CoM.

However, it is only possible for the CoM to accelerate negatively (downwards due to gravity) as joints flex, i.e. knee flexion and ankle dorsiflexion. To achieve this, the jumper reduces agonist muscle activation and/or actively engages antagonist muscles, allowing knee flexion (figure 3) and ankle dorsiflexion under the acceleration of gravity [32]. Initiation of joint flexion in this study is defined as either the start of (concave or convex) curvature, or an inflection point, in the respective knee or ankle angle-time signals derived from motion analysis (figure 3), whichever is earlier. This point of first joint flexion, i.e. first instance of movement, is accompanied by an imperceptibly small decrease in GRF$_z$,

**Table 2.** Phase, sub-phases and events definitions for the countermovement jump (CMJ). Note: Minimum (minima) and maximum (maxima) are the smallest and largest values, respectively, of the signal within a specific interval and/phase (local) or the entire jump duration (global).

| phases of jump | events/sub-phases | traditional definition | new definition |
|---|---|---|---|
| **Downward** (*countermovement*) | 1) Start of CMJ/first instance of movement | When $GRF_z$ decreases by an amount determined either as a (i) percentage, (ii) standard deviation (e.g. 3 times of standard deviation), or (iii) fixed amount (e.g. 10 N) | An inflection point (increase in yank) typically leading (continuously) to the first significant local maximum on the yank-time signal (Event 2) |
| | 2) Meaningful CoM downwards movement | None available directly from force-time signal. Typically, only recognized through the triple integration of the force-time signal and calculating the displacement-time signal. | Significant local maximum of the yank-time signal |
| | 3) Start of braking (deceleration) sub-phase | When $GRF_z$ has increased and equivalent to bodyweight force | When $GRF_z$ has increased to bodyweight force/No definition possible using yank-time signal |
| | 4) Peak braking phase | Not directly available from force-time signal. Peak braking phase can be identified from the peak rate of force development (RFD) using several methods, (i) varying time interval, or (ii) instantaneous RFD (1 millisecond). | Most significant local maximum on the yank-time signal after Event 3. Instantaneous RFD |
| | 5) Lowest height of CoM/ transition point | Highest $GRF_z$ or instant just prior to highest $GRF_z$ during the jump | Second occasion that velocity = 0/Local minimum on the yank-time signal after Event 4 |
| **Upward** (*propulsive*) | 6a) Early upward sub-phase (first 50%) | n.a. | 0–50% of time from Event 5 to Event 7 |
| | 6b) Late upward sub-phase (second 50%) | n.a. | 51–100% of time from Event 5 to Event 7 |
| | 7) End of upward/propulsive phase—last contact with force platform | Instant just before $GRF_z = 0$ N | Instant just before $GRF_z = 0$ N |
| **Flight and landing** | 8) Time in the air (flight phase) | $GRF_z = 0$ N | Time in the air; global minimum of the entire $GRF_z$ |
| | 9) Landing | $GRF_z > 0$ N | $GRF_z > 0$, followed by global maximum of entire $GRF_z$ |

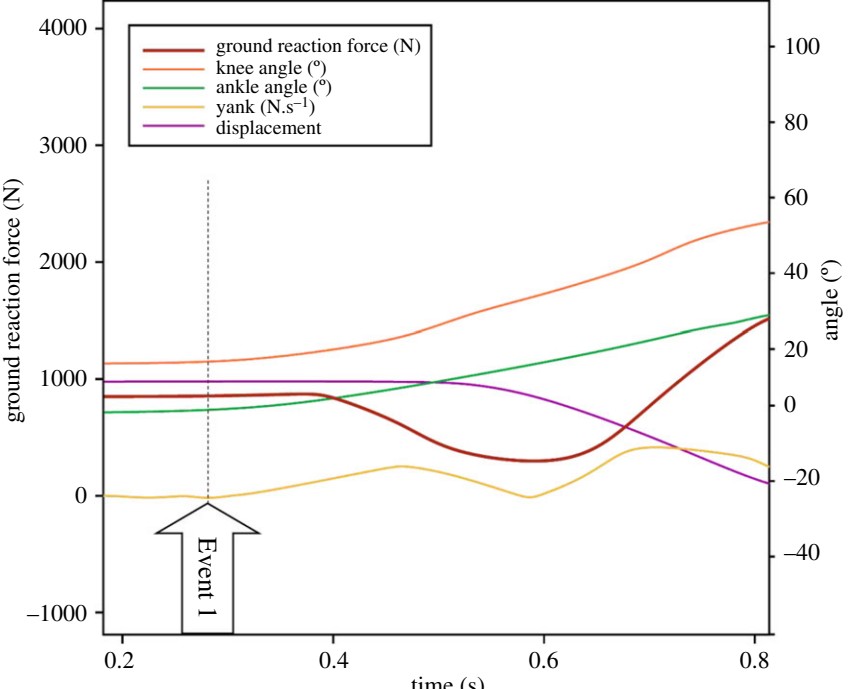

**Figure 3.** Typical temporal changes in knee and ankle joint angles and the vertical ground reaction force (GRF$_z$). Ankle dorsiflexion and knee flexion, i.e. joint flexion (orange and green lines), due to the start of CMJ are reflected as an inflection point (black dotted line) and recognized as Event 1 on the yank-time signal (yellow line). Note that GRFz (brown line) does not noticeably decrease until approximately 0.128 s after this point.

which is barely visually observable and cannot be detected by the methods mentioned above (figure 3). A visual inspection of the initiation of joint flexion and the point of decrease in GRF$_z$ for each subject in the current study revealed that the initiation of the first joint flexion (either ankle or knee) occurred $75 \pm 88$ ms before a significant decrease in GRF$_z$ was first observed using existing methods, while the occurrence of both ankle dorsiflexion and knee flexion occurred $47 \pm 91$ ms before significant GRF$_z$ decrease. Therefore, while Event 1 should be defined as the point where both knee flexion and ankle dorsiflexion has occurred, current methods (table 4) that define Event 1 as a point of significant decrease in GRF$_z$ (e.g. by a percentage of baseline GRF$_z$) are unable to correctly detect the start of the jump (start of downward phase; Event 1). Nonetheless, the small decrease in GRF$_z$ is reflected as a point of positive inflection of the rectified yank-time signal, which typically precedes the first significant local maximum, as shown in figure 3. Therefore, in the present study it is proposed that Event 1 can be identified as the point of positive inflection that leads to the first significant local maximum on the rectified yank-time signal.

To assess the temporal correspondence of Event 1 identified using the yank-time signal relative to previously described methods (described under 'traditional definition' in table 2), the jump start time was calculated using each method (table 2) and then compared with the time at which both ankle dorsiflexion and knee flexion occurred using motion analysis (i.e. direct method). The yank, 97.5% of bodyweight (BW97.5), i.e. where the GRF$_z$ decreases to 97.5% of the bodyweight and the two standard deviation (2SD) methods, i.e. where GRF$_z$ decline exceeds two standard deviations of the baseline GRF$_z$ mean and does not return for at least 50 ms (table 3), provided the closest onset times and smallest bias to those determined directly by motion analysis. However both the Bland–Altman method and regression comparisons of the outcomes of yank and 2SD methods against the motion analysis-derived start times revealed that the limits of agreement for both the BW97.5% and 2SD are large and not a very reliable method of detecting start of the CMJ (Event 1).

Another issue with both these methods are that its reliability is dependent upon the signal-to-noise ratio. These methods are, therefore, likely to be less accurate when the signal contains more noise, which may explain why larger thresholds such 3SD and 5SD have been recommended previously [15]. Regardless, based on the current evidence, the yank-time signal can be used to accurately identify the start of the CMJ (Event 1).

**Table 3.** Methods for determining the time of start of jump (s), and limits of agreement with motion analysis-derived measurements.

| method | bias (s) | | upper limits of agreement | | lower limits of agreement | |
|---|---|---|---|---|---|---|
| | mean | s.d. | upper 95% CI | lower 95% CI | upper 95% CI | lower 95% CI |
| yank | −0.005 | 0.013 | 0.031 | 0.013 | −0.023 | −0.040 |
| percentage of bodyweight (BW97.5%) | −0.41 | 0.678 | 1.709 | 0.865 | −0.948 | −1.793 |
| percentage of bodyweight (BW95%) | −0.138 | 0.121 | 1.643 | 0.784 | −1.060 | −1.919 |
| percentage of bodyweight (BW92.5%) | −0.136 | 0.763 | 1.835 | 0.884 | −1.156 | −2.107 |
| 2x standard deviation (2SD) | −0.015 | 0.438 | −1.148 | −0.602 | −0.570 | −1.116 |
| 3x standard deviation (3SD) | −0.175 | 0.632 | 1.456 | 0.669 | −1.021 | −1.808 |
| 4x standard deviation (4SD) | −0.233 | 0.618 | 1.362 | 0.592 | −1.060 | −1.830 |
| 5x standard deviation (5SD) | −0.106 | 1.170 | 2.917 | 1.459 | −2.401 | −1.672 |

## 4.2. Event 2: first meaningful centre of mass (CoM) movement

As the amplitude of joint flexion increases during the downward phase, $GRF_z$ increases negatively due to gravitational acceleration. This point where $GRF_z$ decreases is traditionally identified as the point where the CoM begins its downwards movement. However, the current data show this to be incorrect; which has been observed in other studies although the authors might not have specifically highlighted it [9,33]. In the present study, a small vertical displacement (less than 10 mm) of the hip joint and body's CoM, defined as the start of a concave downwards point of the CoM vertical displacement-time signal (figures 2 and 3), was detected in some subjects only after $GRF_z$ decreased by at least 34%, and in some cases by as much as 67% depending on the velocity of descent. Temporally this translates to about $81 \pm 78$ ms after a decrease in $GRF_z$ (visual inspection). Currently, the only method available to detect the instance of first CoM movement without direct motion analysis is to inspect the displacement-time data ($d_{CoM}$) derived from the $GRF_z$-time signal. This has been found to be accurate when compared with three-dimensional motion analysis data [33]. Therefore, in the present study, it is proposed that the yank-time signal can be used to identify the first instance of significant CoM movement (Event 2). This event can be recognized as the first most significant local maximum of the yank-time signal, as shown in figure 2.

To determine the temporal accuracy of the yank method to identify Event 2, the timing of the first instance of CoM movement was calculated and visually identified using (i) the displacement-time ($d_{CoM}$) signal, and (ii) the yank-time signal, and then compared with motion capture data (i.e. direct method). The displacement method was found to have a very low level of bias ($d_{CoM} = 4 \pm 13$ ms) and excellent level of agreement when compared with motion analysis-derived data ($d_{CoM} = 0$–9 ms), in accordance with the hypothesis and affirming the observation of an earlier study [33]. The yank method showed a comparable level of bias and level of agreement with motion analysis (bias = $3 \pm 21$ ms, level of agreement = −4–11 ms). Both methods are thus comparable; however, the yank method (identification of Event 2, on the yank-time signal) might be preferred as it allows both Events 1 and 2 to be obtained from a single data manipulation.

## 4.3. Event 3: start of braking (deceleration) sub-phase

The next event, Event 3, marks the start of the braking (deceleration) sub-phase. Traditionally, this event has been defined as the point at which $GRF_z$ is equal to bodyweight [7,15], and thus the acceleration of the CoM is zero. Because this point is easy to define, there is no need to interrogate the yank-time signal to locate it, and in fact this point is not easily definable using yank-time data.

## 4.4. Event 4: peak yank (braking phase)

Event 4 marks the peak rate of $GRF_z$ production, i.e. the greatest positive slope in the $GRF_z$-time relation [34]. It is also sometimes called the peak braking rate of force development (RFD) since it occurs in the

**Table 4.** Definitions of jump profiles based on GRF$_z$ force-time characteristics during the upwards phase.

| jump profile | force-time signal | | rectified yank-time signal | | subjects (n) |
|---|---|---|---|---|---|
| | primary criteria (local maximum/peak) | secondary criteria (local minimum) | primary criteria | secondary criteria (local minimum) | |
| unimodal | only one peak/maximum | n.a. | less than three local minima during the upwards phase + local minimum may NOT necessarily = 0 | n.a. | 12 |
| bimodal-primary[a] | two force maxima, where first maximum ≥ second maximum | local minimum ≥2.5% | three local minima during the upward phase + local minimum = 0 | two local maxima where first maximum ≥ second maximum | 14 |
| bimodal-secondary | two force maxima, where first maximum < second maximum by 10% | | three local minima during the upward phase + local minimum = 0, | two local maxima, where first maximum < second maximum | 6 |

[a]Bimodal-primary is the default profile for all bimodal profiles unless it meets the criteria for bimodal-secondary.

downward (eccentric) phase as the body undergoes negative acceleration. This point coincides with a local maximum in the yank-time signal, which reflects the point at which the rate of change in the slope of the $GRF_z$-time relation first changes from positive to negative. Thus, the yank-time signal can be readily used to identify the point of maximum (peak) rate of $GRF_z$ production against the ground, and in fact the numerical value of the yank-time relation at this point gives the slope of the $GRF_z$-time relation, i.e. instantaneous RFD at any point of time during the vertical jump, without the need for additional calculations as currently practised [31,35].

## 4.5. Event 5: transition point: lowest point of the body's centre of mass (CoM)

The point of lowest CoM occurs at the transition from braking (downwards movement) to propulsion (start of upwards movement). Traditionally, this event has been identified as the instant immediately prior to the peak $GRF_z$ or at the peak $GRF_z$ itself [9]. Other methods used to determine this event include (i) the point at which the positive impulse measured during the rise of force from bodyweight (i.e. zero) force equals the negative impulse generated from the start of the jump to bodyweight (i.e. zero) force [36], and (ii) the instant where CoM velocity is equal to zero [9], since it reflects the point of change of direction of the CoM, which is where negative displacement is the greatest (figure 2). This transition point (Event 5) should be reflected on the rectified yank-time signal as a minimum after Event 4 (peak yank) (figure 2) and can be easily identified in most jump profiles such as in figure 2, but not all, and will be discussed in greater detail in the next section.

To determine the temporal accuracy of the three methods, the transition point (Event 5) was determined using all three methods and compared with motion analysis-derived measurements. Using either the impulse or velocity methods when the value is zero, i.e. velocity is equal to zero, and net impulse is equal to zero, Event 5 was easily identifiable and found to be almost identical (within measurement error), with very low levels of bias (impulse = $3 \pm 18$ ms; velocity = $2 \pm 8$ ms) and both, therefore, provide accurate identification of Event 5. The identification of Event 5 was only easily and clearly identifiable through the yank method as a minimum on 26 subjects and in these cases, had a similar level of bias ($-6 \pm 21$ ms, level of agreement = $-13$ to 1 ms) when compared with motion capture. In the remaining six subjects, the jump profiles were typically unimodal, the minimum indicating Event 5 might be slightly harder to distinguish at first glance as there might be other minima within the same vicinity. However, when the minimum indicating Event 5 is identified correctly, it will have a similar level of bias compared with other methods when compared with motion capture. Based on the findings, the yank-time method is a valid method for the identification of Event 5, though it is acknowledged that the identification of Event 5 for some jump profiles might be harder using the yank-time method and therefore the impulse method is recommended as it can be easily calculated directly from the original GRFz-time signal regardless of jump types without the need to calculate the velocity-time signal.

On the contrary, in 12 subjects, Event 5 occurred up to 400 ms ($197 \pm 104$ ms) before peak $GRF_z$ (figure 4, right), in eight subjects there was little or no difference between the lowest point of the CoM and peak $GRF_z$ ($8 \pm 10$ ms), and in the remaining 12 subjects the lowest point of the CoM occurred up to 96 ms ($23 \pm 33$ ms) after the first local maximum of the $GRF_z$-time signal. Therefore, the use of peak $GRF_z$ to identify the lowest point of the CoM is not valid. The velocity method was subsequently used in the present study to identify the lowest point of the CoM due to its simple calculation and lowest level of bias.

## 4.6. Interpreting the shape of $GRF_z$ during the upwards phase of a CMJ with a self-selected countermovement depth

Event 5, the transition point, also signals the start of the upward phase, which lasts until Event 7 when the jumper loses contact with the force platform and becomes airborne. In addition to the calculation of mechanical variables, the shape of the $GRF_z$-time relation during the upward phase may provide further insight into lower-body function, as this shape has been shown to differ between individuals and to change after physical activity interventions such as strength and power training [37,38]. The force-time relation during the upward phase can be divided into two sub-phases or periods: early (first 50% of the upwards phase, Event 6a) and late (remaining 50% of the upwards phase, Event 6b). Depending on whether a visible reduction (fall) and subsequent increase (rise) in $GRF_z$ occurs during the upward phase, the $GRF_z$-time relation can be classified as either:

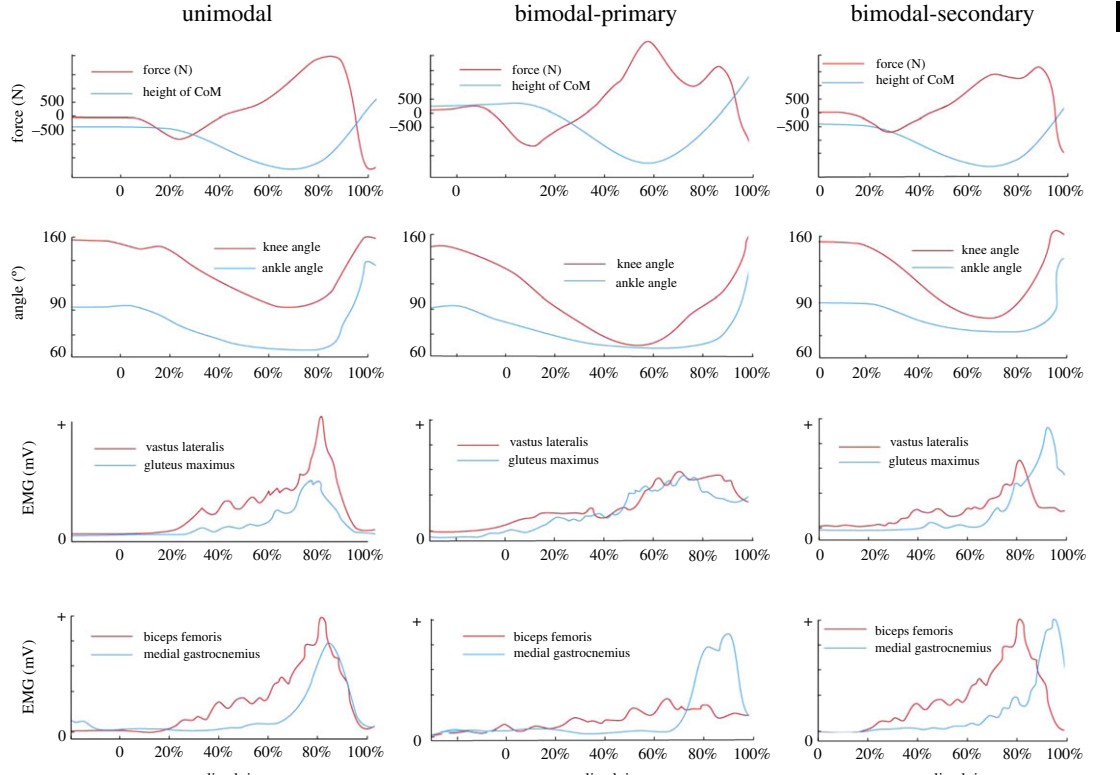

**Figure 4.** Jump profiles based on GRF$_z$-time signal shape, knee angles and EMG activity of the muscles within the cohort. Unimodal (left; one force peak), bimodal-primary (middle; two force peaks, first peak is higher) and bimodal-secondary (right; two force peaks, second peak is higher). The height of CoM (first row, blue line) starts from zero. The second row shows the knee and ankle angle while the third and fourth rows show the EMG activities of the respective muscles.

(1) unimodal: GRF$_z$ rises to a local maximum then falls to zero without any other significant rises or falls, e.g. figure 4 (top, left);

(2) bimodal-primary: GRF$_z$ rises to a local maximum during the first sub-phase (Event 6a; described below) followed by a reduction (i.e. local minimum) and subsequent rise (local maximum) during the second sub-phase (Event 6b; described below), with the maximum force occurring *before* the secondary fall, e.g. figure 4 (top, middle);

(3) bimodal-secondary: GRF$_z$ rises to a local maximum during the first sub-phase (Event 6a) followed by a reduction (local minimum) and subsequent rise (local maximum) in GRF$_z$ during the second sub-phase (Event 6b), with the maximum force occurring *after* the secondary reduction, e.g. figure 4 (top, right). While these jump profiles can be determined by examination of GRF$_z$-time signals, they can also easily be determined from the yank-time signal, since local minima occur in the rectified signal at each turning point (beginning of each rise or fall) of the GRF$_z$-time signal (see figure 2 for example).

Based on the shape of the two best jump trials or the effective majority (if they performed more than three jumps), the subjects were categorized into one of three groups (table 4). While the possibility has been suggested that there might be a within-subject inconsistency in the shape of the GRF$_z$-time signal [39], no such inconsistencies were found in the present study between the best two (or the effective majority of) self-selected jump trials in the current cohort of subjects. It was then determined whether their profiles were associated with a specific movement pattern that could be used to infer the muscle activation strategies of an individual. Examination of the differences between the three groups revealed that the propulsive phase of unimodal jumpers was significantly shorter in duration than in bimodal jumpers, while bimodal-primary jumpers tended to spend more time in the propulsive phase.

### 4.6.1. Differences in the yank-time signal during the upwards phase across the different profiles

It was observed that the different profiles could be differentiated with greater clarity through interrogation of the yank-time signal characteristics (table 4), especially in situations where the difference between the two local maxima was small, or a momentary plateau appeared as a local

minimum and might thus be classified as a unimodal or bimodal-secondary profile. In a unimodal profile, the yank-time signal will not have a minimum point during the upwards phase that is equal to zero, unlike in the bimodal profile where there is a clear local minimum that equates to zero. The two local $GRF_z$ maxima are magnified in the yank-time signal, allowing for easier distinction between bimodal-primary and bimodal-secondary profiles.

## 4.7. Muscle activity: EMG onset

The onset of agonist muscle activity ($EMG_{on}$) should not occur before the start of the CMJ (Event 1) because the jumper first has to decrease muscle activity, or activate antagonist muscles (figure 4, rows 3 and 4), in order to allow the body to fall with gravity. Once CoM movement starts (Event 1), the agonist muscles must then activate to produce forces sufficient to decelerate the CoM prior to the lowest CoM height (Event 3). To assist in identifying the point at which force first starts to be produced, the earliest $EMG_{on}$ of agonist propulsive muscles (see below) was plotted against the (i) start of jump (Event 1), (ii) first CoM movement (Event 2), and (iii) the smallest $GRF_z$ magnitude (between Events 2 and 3), to observe the limits of agreement and relationships between the different methods. This was used to ascertain their suitability as a point associated with the first activation of muscles ($EMG_{on}$) and subsequent change in the slope of the $GRF_z$-time relation.

The earliest $EMG_{on}$ of an agonist muscle in either the hip (e.g. gluteus maximus and/or biceps femoris in most subjects) or the knee (e.g. vastus lateralis or vastus medialis in other subjects) occurred, on average $98.1 \pm 37.9$ ms after Event 1 (start of the jump) or from another perspective about $3 \pm 19$ ms before Event 2 (significant CoM movement). It is important to note that there is always some error in the determination of the true point of onset of muscle activation, and it is very likely that true onsets occurred prior to the observed $EMG_{on}$ in the current study. Nevertheless, the observation that $EMG_{on}$ of at least one agonist muscle occurred closest and in most cases prior to Event 2 suggests that the muscles were activated and force produced in the vicinity of this event (Event 2) explaining the change in rate of force; as expected, $EMG_{on}$ was slightly in advance of Event 2 because there will be an electromechanical delay [40–42]. Therefore, Event 2 can be considered as the point where external $GRF_z$ might be considered to occur.

## 4.8. Muscle activity: peak EMG ($EMG_{peak}$)

In general, the sequential timing of peak EMG ($EMG_{peak}$) in the agonist muscles revealed a proximal-to-distal pattern (table 5). In unimodal jumpers, $EMG_{peak}$ amplitudes for all agonist muscles were reached within 80 ms prior to the maximum $GRF_z$ (figure 4, top). For the bimodal jumpers, $EMG_{peak}$ activities were more temporally separated between muscles (by as much as 287 ms, between the first and last $EMG_{peak}$), occurring around the time at which the two $GRF_z$ maxima were obtained (figure 4, middle and bottom); for bimodal-primary the peaks were mainly reached at or before the first peak, and for bimodal-secondary the peaks were largely reached at or before the second peak. It is notable that peak medial gastrocnemius EMG activity was reached near the second maximum of the $GRF_z$-time data for all bimodal-secondary jumpers. The late medial gastrocnemius $EMG_{on}$ and $EMG_{peak}$ suggests that activation of, and force production by, this muscle might have significantly contributed to the presence of a second, higher maximum $GRF_z$ in these individuals with a bimodal-secondary profile.

## 4.9. Muscle activity: medial gastrocnemius

In all subjects, medial gastrocnemius $EMG_{on}$ occurred later in the jump than for other muscles, at 48–56% of the total jump time (table 5). Of note in bimodal-secondary jumpers was that while $EMG_{on}$ occurred late, approximately 56% into the jump, the peak occurred soon after, at approximately 83% into the jump, i.e. the activation rate appeared to be rapid for medial gastrocnemius in these jumpers. The temporal EMG patterns for medial gastrocnemius reinforces the suggestion that it plays a role in the generation of a second local maximum of $GRF_z$ during the upwards phase of the CMJ.

## 4.10. Joint kinematics

Three different joint kinematic patterns were observed in the upward phase of the CMJ, with the main difference relating to the timing of knee extension and ankle plantar flexion onsets (figure 4). The first movement pattern, which was exhibited by all unimodal jumpers, was associated with a short time

**Table 5.** Timing of minimum and maximum GRF$_z$, lowest height of CoM, and time of onset and peak EMG as a percentage of jump time (0 to 100%) for the different jump profiles.

| jump profiles | | first local minimum on GRF$_z$ | E5[a] | first local maximum on GRF$_z$ | second local maximum GRF$_z$ | erector spinae | | gluteus maximus | | vastus lateralis | | vastus medialis | | biceps femoris | | medial gastrocnemius | | soleus | | anterior tibialis | |
|---|---|---|---|---|---|---|---|---|---|---|---|---|---|---|---|---|---|---|---|---|---|
| | | | | | | onset | peak | onset | peak | onset | peak | onset | peak | onset | peak | onset | peak | onset | peak | onset | peak |
| unimodal | mean | 24.9 | 68.9 | 84.9 | — | 18.2 | 78.2 | 23.1 | 83.6 | 20.6 | 85.3 | 22.9 | 79.4 | 20 | 79.5 | 50.1 | 87.5 | 46.9 | 80.3 | 13 | 58.6 |
| | s.d. | 8.3 | 3.6 | 8.2 | — | 12.2 | 7 | 8.3 | 6.2 | 11.7 | 7 | 18.1 | 24 | 17.9 | 3.5 | 11.8 | 3.2 | 17.6 | 10.1 | 7.8 | 42 |
| bimodal-primary | mean | 18.6 | 64 | 65.6 | 89.1 | 11.7 | 70.2 | 17.9 | 80.1 | 16.6 | 83.5 | 13.1 | 83.4 | 18.7 | 77.2 | 48.5 | 89.1 | 39.9 | 82.6 | 16 | 38.1 |
| | s.d. | 7.9 | 3.9 | 6.8 | 2.4 | 6.3 | 21.6 | 4.4 | 7.2 | 9.4 | 8.4 | 7.8 | 10 | 13.7 | 11 | 14.5 | 4.6 | 12.7 | 4.7 | 31.5 | 17.4 |
| bimodal-secondary | mean | 21.5 | 64.3 | 66.6 | 88.8 | 14.8 | 71.5 | 19.2 | 86 | 21.4 | 85.5 | 26.9 | 87.4 | 22.9 | 77.4 | 56.2 | 83.2 | 62.8 | 83.9 | 22.6 | 46.7 |
| | s.d. | 4.2 | 4 | 3.8 | 2.6 | 3.9 | 11 | 3.9 | 6.2 | 7.9 | 5.4 | 8.7 | 8.1 | 13.2 | 6.7 | 16.2 | 5.7 | 9.5 | 3.8 | 17.3 | 29.7 |

[a]Event 5, lowest height of CoM.

(less than 35 ms) between initiation of extension of these joints, i.e. they were near-simultaneous (figure 4, joint angle–unimodal). Bimodal jumpers, however, exhibited either of two remaining patterns. One (figure 4, joint angle–bimodal-primary) was observed in 4 of 15 bimodal-primary jumpers but 5 of 6 bimodal-secondary jumpers, and was associated with a delay in the commencement of ankle plantar flexion (more than 94 ms after knee extension began), typically during the local minimum between the two local maxima on the force-time curve during the upwards phase. In the remaining bimodal jumpers (figure 4, joint angle–bimodal-secondary), ankle plantarflexion began soon after the commencement of knee extension; however, plantarflexion was initially very slow and began to increase notably in speed after approximately 89 ms.

# 5. Discussion

In the present study, the countermovement vertical jump (CMJ) was used as a model in which to examine the use of yank-time data to infer kinematic, kinetic and muscle activation patterns in complex, multi-joint movement tasks. By using the yank-time signal and comparing it with motion analysis data, it has been possible to identify several key events. These include the first instance of lower-limb joint flexion (i.e. start of the jump; Event 1), which is often missed, the point at which the first CoM movement and (probably) onset of force occur (Event 2), the point of the maximum (peak) (Event 4), and the lowest point of the CoM, i.e. downward–upward phase transition point (Event 5). It was also possible from examination of either $GRF_z$-time or yank-time signals to differentiate three main $GRF_z$-time profiles, i.e. unimodal versus bimodal-primary versus bimodal-secondary, which were found to be associated with characteristic kinematic patterns and to be underpinned by specific muscle activation patterns. Thus, from a detailed analysis of $GRF_z$-time data, particularly after derivation of the yank-time signal, it was possible to accurately infer both movement (kinematic) and muscle activation patterns in humans performing a complex, multi-joint movement task. Based on these findings, the possibility to identify important kinematic, kinetic and muscle activation events during complex human (or other animal) motion through interrogation of yank-time data appears to be a realistic goal and can be used as an alternative to existing methods without the need to compare with cohort data.

## 5.1. Use of the yank-time signal to define CMJ events and phases of the countermovement jump (CMJ)

The interrogation of the yank-time signal has revealed several kinematic events that cannot be accurately detected in the CMJ by examination of force-time data alone, and it has not yet been possible to link specific events to underlying muscle activation patterns. As a first example, the point at which the lower-limb joints first flex, and thus the CMJ starts, has not been accurately identified using vertical ground reaction force ($GRF_z$)-time data alone (e.g. figures 2 and 3); instead, motion analysis has had to be introduced. This point is important because it defines the point at which analysis of the movement must commence. In the yank-time data, CMJ start was identifiable as a point of positive inflection of the rectified yank-time signal, which typically precedes the first significant local maximum as shown in figure 2. This point was found to occur $75 \pm 88$ ms before a significant decrease in $GRF_z$ was first observed using existing methods, which is the point previously used to identify the jump start, indicating the significant error that can be made when $GRF_z$-time data are used to identify the start of the jump. Thus, interrogation of yank-time data allowed for the accurate identification of a kinematic event that is typically too subtle to be accurately identified from $GRF_z$-time data alone.

In fact, the start of the CMJ (Event 1, joint flexion) occurs slightly before the first detectable decrease in height of the body's centre of mass (CoM) is observed, i.e. Event 2. Event 2 can be recognized as the first most significant local maximum of the yank-time signal, as shown in figure 2. However, in previous studies the first movement of the CoM was assumed to represent the start of jump and identified from $GRF_z$ data as (i) the point at which $GRF_z$ first starts to fall by a given multiple of the standard deviation of the baseline signal [29,30], (ii) a decrease in $GRF_z$ as a percentage of bodyweight [15], or (iii) an absolute change in $GRF_z$, e.g. when $GRF_z$ decreases by 10 N [31]. However, these methods were found to locate the start of the jump (Event 1) either too early or significantly later, with an average error of 163 ms (i.e. approx. 20% of total CMJ time), whilst a close temporal alignment was observed between Event 2 identified from the first maximum of the rectified yank-time data and the first meaningful movement of the CoM detected by motion analysis (mean error versus motion analysis = 7 ms). Therefore, indirect methods that use the $GRF_z$-time signal tend to be inaccurate; not

only do they not identify the true start of the CMJ (Event 1) but they do not correctly identify the start of meaningful CoM movement (Event 2) or any other important kinematic or kinetic event. The rectified yank-time signal has proven useful, however, since the first inflection point aligns with the start of the jump (Event 1, joint flexion) while the local maximum at this point in the signal aligns with the first movement of the CoM (Event 2; figure 2).

It is notable that Event 2 also occurs soon after the first onset of muscle activity $EMG_{on}$ of one or more proximal agonist muscles (usually gluteus maximus or vastus lateralis) and it appears at a point at which the slope of the $GRF_z$-time relation first starts to tend less negatively, signifying a change in $GRF_z$ production (figure 2). Therefore, this newly defined Event 2 serves as a close approximation of $EMG_{on}$ and force production in the CMJ, after the preceding decrease in both muscle activity force production. Event 2 might also therefore serve as a more correct starting point for the calculation of other variables such as rate of force development, time to peak force, modified reactive strength index, and others, rather than use of the point of first decrease in $GRF_z$.

While Event 3 (start of the braking [deceleration] sub-phase) is easy to locate as the point at which $GRF_z$ is equal to bodyweight [15,7], it is not easily definable using yank-time data.

Event 4 marks the peak yank or rate of $GRF_z$ production (RFD; rate of force development), i.e. the greatest positive slope in the $GRF_z$-time relation. This peak yank has also been referred to as the peak braking RFD since it occurs in the downward (eccentric) phase as the body undergoes negative acceleration. Event 4 coincides with a local maximum in the yank-time signal, where the rate of change in the slope of the $GRF_z$-time relation first changes from positive to negative. Because it is an easily identifiable point in the yank-time signal, and the numerical yank value at this point gives the slope of the $GRF_z$-time relation, the peak RFD can be taken as the point of maximum (peak) rate of $GRF_z$ production against the ground. In fact, any point on the yank-time signal gives RFD at that point in time.

The correct identification of the point of lowest CoM (Event 5) is also an important event, which has traditionally been identified as the point at which the peak ground reaction force, $GRF_z$, occurs during the jump. This makes theoretical sense, but was found only to be true in a small subset of jumpers who exhibited a bimodal propulsive force profile (i.e. with two distinct propulsive force peaks; 8 of 32 in the present study). This was problematic in jumpers with unimodal profiles in particular, where the point of lowest CoM height occurred up to 255 ms ($177 \pm 63$ ms) before the point of maximum $GRF_z$. The important conclusions from this part of the analysis, therefore, are that (i) the point of lowest CoM height cannot be assumed to occur at, or near, the peak $GRF_z$, and (ii) instead, the point of lowest CoM height should be obtained from the velocity-time signal, integrated from the force-time signal, where velocity is equal to zero. The impulse method (where nett impulse is equal to zero) is recommended for the identification of Event 5 as it can be easily calculated directly from the $GRF_z$-time signal regardless of jump profiles and without the need to derive the velocity-time signal. Similarly, Event 5 can be easily identified using the yank-time method for most jump profiles with the exception of a unimodal-secondary jump profile (figure 5, left). While it will still be reflected as a minimum, it may not be as distinct as in the other jump profiles (figure 4, middle and right).

## 5.2. Interpreting the unimodal GRFz profile

Individuals with a unimodal force profile might be expected to exhibit a similar (classic) CMJ strategy (technique). However, further analysis of individuals who displayed a unimodal $GRF_z$ profile suggested that, in fact, two jumping profiles were apparent: unimodal-secondary and unimodal-primary (figure 5). The key differentiator is the location of the points of lowest height of the CoM (Event 5) and both peak $GRF_z$ (tables 5 and 6) and EMG activity (i.e. $EMG_{peak}$). In half (6 out of 12) of the unimodal jumpers, the point of lowest CoM height (Event 5) occurred up to 251 ms ($139 \pm 101$ ms) or $20.2 \pm 5.1\%$ of normalized time before the point of maximum $GRF_z$; as their lowest CoM occurred before $GRF_z$ maximum, they were labelled as unimodal-secondary (figure 5, unimodal-secondary). Also of note in these jumpers was that agonist muscle EMG amplitudes at the point of lowest height (Event 5) were less than $22 \pm 9\%$ of the peak EMG amplitudes. That is, the muscles had not reached near-maximal activity, and the force at the point of lowest CoM was not maximal. As expected, peak EMG activity occurred less than 80 ms before the point of maximum $GRF_z$, which occurred later in the jump; i.e. the subsequent rise in $GRF_z$ (a further $15.4 \pm 4.7\%$ of CMJ time) after the point of lowest CoM height resulted from continued increases in muscle activity. In these jumpers, the muscles achieved their peak activities either simultaneously or in very close temporal proximity (figure 4, left). This simultaneous and rapid increase in muscle activity ensured that a secondary reduction then further increase in $GRF_z$ could not occur and prevented a bimodal jump profile from being exhibited.

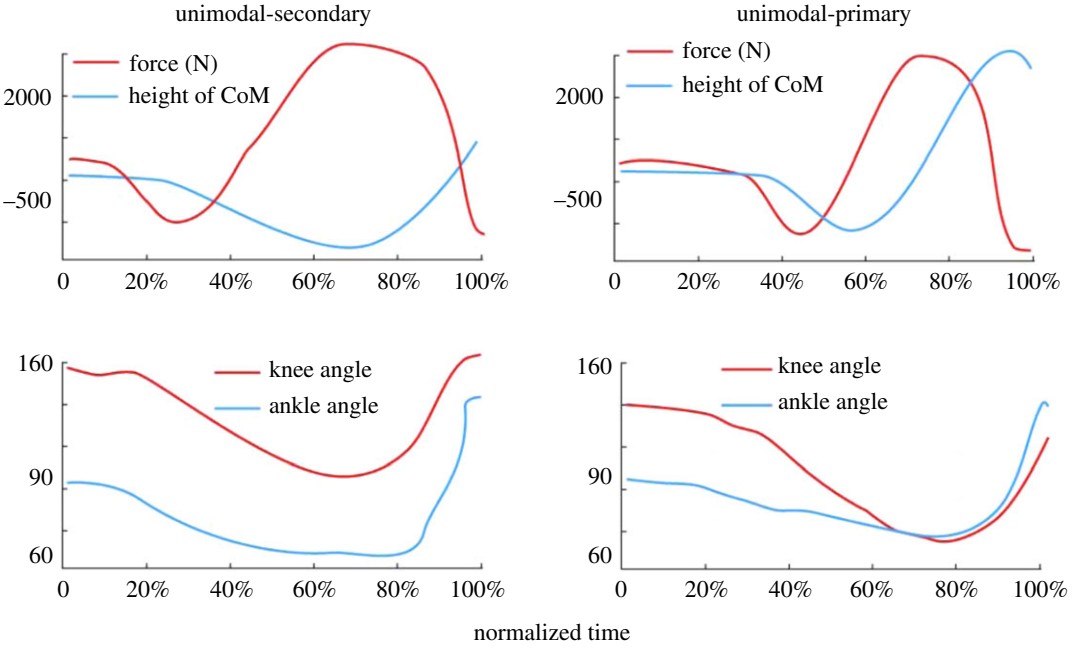

**Figure 5.** Difference between the two different types of unimodal jump profiles. In a unimodal-secondary jump profile, the point of lowest height of CoM (Event 5) occurred before the point of maximum $GRF_z$. In a unimodal-primary jump profile, the point of lowest height of CoM (Event 5) occurred close to or at the point of maximum $GRF_z$.

**Table 6.** Definitions for the four jump profiles.

| | force-time signal | |
|---|---|---|
| jump profile | primary criteria (local maximum/peak) | secondary criteria (local minimum) |
| unimodal-primary[a] | only one maximum/peak | there is less than 10% difference in normalized time between Event 5 and point of maximum $GRF_z$ |
| unimodal-secondary | | there is a more than 10% difference in normalized time between Event 5 and point of maximum $GRF_z$ |
| bimodal-primary[a] | two force maxima/peaks with a local minimum greater than or equal to 2.5% between the two force maxima | first force maximum greater than or equal to second force maximum |
| bimodal-secondary | | two force maxima, where first maximum is less than second maximum by 10% |

[a]The primary (both unimodal and bimodal) are the default profiles unless the specific secondary criteria are also met.

In the remaining unimodal jumpers, i.e. unimodal-primary (figure 5, unimodal-primary), the point of lowest CoM height (Event 5) occurred close to or at the point of maximum $GRF_z$ (mean difference = 81.8 ± 80.65 ms or 3.6 ± 2.5% of normalized time). In these unimodal-primary jumpers, the lowest CoM height occurred at or after peak EMG activity, and the resulting peak force had therefore also occurred prior to the lowest height of CoM being reached. Unlike the unimodal-secondary jumpers, there was little or no additional increase in $GRF_z$ as the CoM rose in the upward phase.

While there is no significant difference in the timing of the point of lowest CoM height (Event 5) between all four of the jump profiles (tables 5 and 6), all occurring at about 64% of the jump, there is a significant difference ($p < 0.01$) in the timing of the point of maximum $GRF_z$ in the unimodal-primary, which occurred at about 71.9 ± 4.5%, when compared with the unimodal-secondary (86.6 ± 3.9%) and bimodal jumpers. This difference suggests that, when compared with the unimodal-secondary jumpers, the unimodal-primary jumpers had a less optimum jumping technique since they

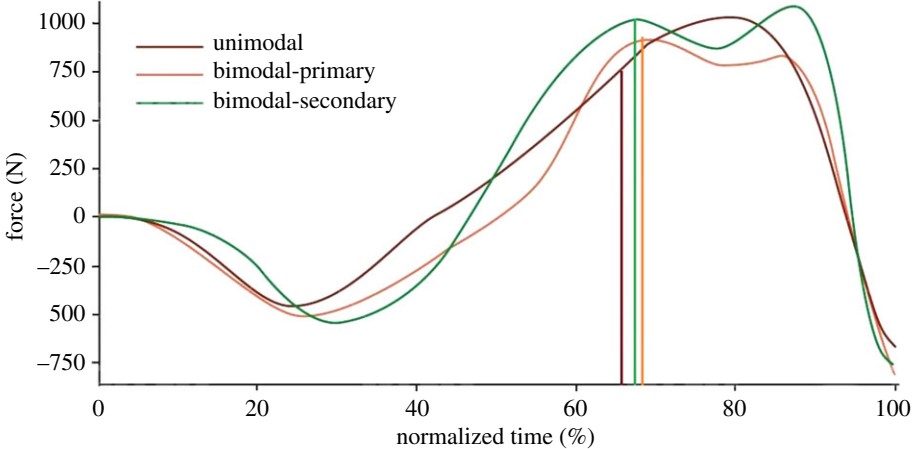

**Figure 6.** Comparison of GRFz-time relations for the three jump profiles normalized to jump time. Unimodal-secondary, bimodal-primary and bimodal-secondary are compared using normalized time. Based on the occurrence of Event 5, the three respective vertical lines in unimodal-secondary and the first local maximum for the bimodal jumpers, we speculate that the activation patterns of bimodal jumpers are similar to the unimodal-secondary jumpers.

did not further increase $GRF_z$ once they commenced the upward (propulsive) phase. Based on these data, it is speculated that unimodal-secondary, but not unimodal-primary, jumpers might have an optimized jumping technique. In line with this, the six unimodal-secondary jumpers were ranked second to sixth and tenth out of the 12 unimodal jumpers for jump height.

## 5.3. Interpreting the bimodal $GRF_z$ profile

In contrast to unimodal jumpers, 20 out of 32 jumpers in the present study exhibited two distinct force peaks during the upwards phase of the jump; they were thus labelled as bimodal jumpers. For these jumpers, temporal separation between knee flexion and ankle plantarflexion after the point of lowest CoM height was observed using motion analysis, indicating a contribution of ankle plantarflexion to the second maximum of the $GRF_z$-time signal. Within these jumpers, the peak agonist EMG amplitudes typically occurred at or just after the first peak in some jumpers (figure 4, middle) while peak EMG amplitudes occurred after the first peak or at the second peak in others (figure 4, right). Based on this criterion, the former were labelled as bimodal-primary, whereas the latter were labelled as bimodal-secondary. These profiles differ from unimodal jumpers, where knee flexion, ankle plantarflexion and peak EMG activities tended to occur almost simultaneously (figure 4, left). The temporal separation of knee and ankle extensions and peak EMG activities after the point of lowest CoM height might possibly explain the presence of the second $GRF_z$ peak in these jumpers.

The point of lowest CoM height (Event 5) almost always occurred at the first local $GRF_z$ maximum. The time difference between this point and the second local maximum in unimodal-secondary jumpers was similar to the time difference between the point of lowest CoM height and peak $GRF_z$. When considered alongside the finding of temporal separation of the knee flexion, ankle plantarflexion and timing of the peak EMG amplitudes, it may be speculated that the activation patterns of bimodal jumpers are somewhat similar to the unimodal-secondary jumpers, when time-normalized data are compared [43] (figure 6).

One significant difference between bimodal-primary and bimodal-secondary jumpers is the late onset and peak medial gastrocnemius EMG activity observed in bimodal-secondary jumpers, which typically occurred at or before the second $GRF_z$ peak. Considering that the ankle joint is a significant contributor to both positive jump power (about 31%) and work (up to 64%) [44], it is possible that this late and strong medial gastrocnemius activation contributed to (i.e. caused) the second, higher $GRF_z$ peak in bimodal-secondary jumpers. Thus, the muscle activation strategies and the resulting kinematic patterns are strongly associated with the $GRF_z$ profile of the jumpers.

The temporal separation between proximal and distal joint extensions and agonist muscle activity observed in bimodal jump profiles (figure 3, middle and right) suggests that it might also be observed in a CMJ with a greater countermovement since a deeper countermovement during the downwards phase will ensure a greater knee flexion. Knee extension must then occur significantly earlier than ankle

plantarflexion to overcome the greater knee flexion and achieve the required joint extension for jumping. Therefore, it is possible that individuals who use a greater countermovement depth during the downwards phase will exhibit a bimodal jump profile. This hypothesis was not found to be true in the current study, as there was no significant difference ($p = 0.05$) in the lowest height of CoM during the downwards phase ($p = 0.55$) or knee angle ($p = 0.47$) between unimodal and bimodal jumpers during a self-selected CMJ. However, it would be of interest to test whether instructing individuals to use a greater countermovement depth, i.e. a different movement pattern in unimodal jumpers, leads to adoption of a bimodal jump profile.

# 6. Conclusion

In the present study, derivation of the $GRF_z$-time signal to yield the yank-time signal allowed for the identification of several kinematic/kinetic events during, and phases of, the CMJ that have either not previously been easily identifiable or have been identified incorrectly from the examination of $GRF_z$-time data alone. In some cases, these events and phases had to be redefined or clarified, which would probably affect variables that depend on, or relate to, them (e.g. rate of force development and the modified reactive strength index). Indeed, the term yank is a more correct term to describe the rate of force development and we encourage its use. Therefore, the yank-time signal offers a viable alternative to identify and describe important kinematic and kinetic events during the CMJ without the need for direct motion capture. Additionally, interrogation of the yank-time signal in conjunction with motion analysis and EMG data allowed for the identification of two broad $GRF_z$-time profiles, each of which can be further differentiated to two further profiles for a total of four different CMJ profiles. Jumpers with these $GRF_z$ profiles were found to exhibit specific kinematic and muscle activation patterns. Thus, it would now be possible to broadly infer kinematic and muscle activation patterns from examination of the yank-time signal (with some information still obtained from the $GRF_z$-time signal) of an individual with reasonable confidence. These data suggest that inference of kinematic and muscle activity patterns from external force production data, at least after derivation, might be possible for other complex human tasks, or movements performed by other animals.

Ethics. Ethical approval for this study has been granted by the ECU Human Research Ethics Committee (HREC), project no. 14378 SAHROM. In granting approval, the HREC has determined that the research project meets the requirements of the National Statement on Ethical Conduct in Human Research.

Data accessibility. All relevant data are within the manuscript and at the public repository https://github.com/GitHub-Sofyan/BJH-yank-time-cmj.

Authors' contribution. S.B.S. and A.J.B. participated, conceived and designed the study and helped draft the manuscript. J.C.W. and K.N. critically revised the manuscript and helped in analysis of the data. S.B.S. carried out the data collection and analysis.

Competing interests. The authors declare that they have no competing interests.

Funding. This research was supported by Edith Cowan University Postgraduate Research Scholarship (International). The funders had no role in study design, data collection and analysis, decision to publish, or preparation of the manuscript.

Acknowledgements. We would like to acknowledge Ms Julia Adorno Fernades and Ms Larrisa Bocarde from Universidade de São Paulo (USP), Mr Cody Wilson, Mr Abdul Matin, Mr Fabian Inyou and Mr Wayne Poon from Edith Cowan University (ECU) for their assistance with data collection.

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
