## [Reviewer comments · Royal Society Open Science]

Review History

RSOS-192093.R0 (Original submission)

Review form: Reviewer 1 (Dale Chapman)

Is the manuscript scientifically sound in its present form?

Yes

Are the interpretations and conclusions justified by the results?

Yes

Is the language acceptable?

Yes

Do you have any ethical concerns with this paper?

No

Have you any concerns about statistical analyses in this paper?

No

Recommendation?

Major revision is needed (please make suggestions in comments)

Comments to the Author(s)

General Comment

The authors have conducted a thoughtful examination of a potential additive signal analysis technique for a deeper understanding of vertebrae locomotion initially using the countermovement jump as the investigated movement. It was not apparent to this reviewer when reading the introduction that this was intended to be a technique used in conjunction with the current practice and I suggest that the authors reconsider the theoretical presentation to ensure the reader understands earlier that use of the Yank-time signal processing is to be used in cohort with other methods to improve the accuracy and reliability of event detection. This fact only becomes apparent in the conclusion. It is potentially outside of the scope for this manuscript but as the authors are seeking to instigate the use of new analytical method presenting the intra-individual reliability of the method for identifying the various events X-axis time variation and the Y-axis magnitude variation from the 3 maximal CMJ performed would aid the reader greatly. It would be of interest to readers to know how this technique other than the ability to identify some events with greater certainty is better than applying statistical techniques such as spatial parametric mapping or functional principle component analysis to the GRF-time signal provides for a deeper understanding of the movement complexities? Granted the Yank-time signal as presented has strong alignment with sEMG onset and peak qualities (within analysis limits) and similarly with positional displacement data which also has it's own variation and limitations due to skin movement artefact etc, however all of these have been previously reported and accepted based on broad percentages of the movement time. Thus even SPA and fPCA all report back to a time normalisation process, so what is the use of Yank truly gaining the practitioner/scientist?

Specific Comment

Ln 23; suggest changing 'allowed' to 'informed' and 'greater' to 'deeper'

Pg 7 Ln 25; Please detail which model was used for the coordinate reconstruction?

Pg 8 Ln 12; Although at least 3 maximal efforts were performed only a CMJ single trial was analysed (selected from jump height), why not use an average of all jumps performed

Pg 11 Figure 2; As the authors are advocating for the use of the term Yank instead of RFD then it would be beneficial to the overall paper if this were consistently applied.

Pg 12 Table 2; Event 1) it is unclear how the new definition is any more accurate and it is arguably less objective than the traditional definition for the start of movement. Define with objective numerical terms what is required for a significant local maximum; Event 4) Considering that this event by the nature of the term used to describe it as Peak Rate of Force Development, it is a misnomer to indicate that it is not possible to integrate the signal for the resultant variable i.e. as has been shown for Event 2, which is then as described in the manuscripts introduction the commonality between Yank and Rate of Force Development (see also Journal of Experimental Biology 2019 222: jeb180414 doi: 10.1242/jeb.180414). Event 6a and 6b, would this better described as 6a: 0-50% of time from Event 5 to Event 7 (toe off) and 6b: 50-100% of time from Event 5 to Event 7 (toe off)?

Pg 14 Ln 4; the authors need to be more precise and provide objective information as to the tended alignment of the yank-time signal to the identifiable GRF-time signal (see general comment on reliability).

Pg 15 Ln 1; should this be referring to Table 2 as Table 4 is not providing movement onset definitions?

Pg 15 Ln 1-7; If there has been no change in the GRF and Yank is the integration of GRF with time, how is there a measurable change in the resultant Yank without it being the result of noise? Did the authors also apply any filtering techniques to the Yank as similarly applied to the other signals?

Pg 22 Ln 1-3; This statement is in contrast with that presented at Pg 8 Ln 12

Pg 22 Ln 3-6; Can the authors please detail the statistical approach they used to determine that there were no with-in subject variation to the shape of the GRF-time signal?

Pg 24 Ln 9; Grammar, should read, 'It is notable that...'

Pg 30 Table 6; check the grammar for the secondary criteria of the Unimodal-primary jump profile

Review form: Reviewer 2 (Drew Harrison)

Is the manuscript scientifically sound in its present form?

Yes

Are the interpretations and conclusions justified by the results?

Yes

Is the language acceptable?

Yes

Do you have any ethical concerns with this paper?

No

Have you any concerns about statistical analyses in this paper?

No

Recommendation?

Accept with minor revision (please list in comments)

Comments to the Author(s)

GENERAL COMMENTS

This paper examines the use of yank-time function as a way to identify kinematic events and define key phases and events in the countermovement jump.

The paper is very well written and the topic is important despite the prevalence of research on countermovement jumping across many years. The study identifies and addresses one of the key problems encountered by researchers in analysis force time data on countermovement jumping, namely, the difficulty in accurately defining key events on the time series such as the start of the countermovement since any error at this point carries forward and propagates errors in the constant of integration which will accumulate during integration. This is not an easy problem to resolve especially since neither the movement data or force data provide any correct value (gold standard) to test the accuracy of the proposed solution. The authors present a comprehensive analysis of how the yank-time function may offer a solution to the identification of the events and phases. The analysis includes kinematic and EMG data well as force time data to allow cross-validation of the effectiveness of the yank-time function. I commend the authors for a very thorough treatment of a challenging problem and I believe the content of the work will have good impact in the field. The manuscript is generally well written and well structured and therefore I only recommend various minor textual changes and clarifications (see specific comments below.

SPECIFIC COMMENTS

P3 ln 2: this is a verbatim repeat of the abstract... maybe some rewording would be appropriate

P 3 ln 9 : Perhaps omit however

P3 ln 25-26: passive expression... perhaps avoid by reversing the sentence clauses: "The vertical countermovement jump (CMJ) is a complex human movement task that has been well studied from a force-time perspective [6, 7]" I encourage the authors to consider similar instances of passive expression throughout the manuscript and reverse the clauses as this will improve the readability of the paper.

P4 Table 1: This table is very helpful in describing the derivative terminology however the reference suggests to me that this may not be an original source ... if this is not the original source can you please use the original source.

P8 In 26: Teiger-Kaiser Energy Operator, the spelling is different in the cited reference. Please correct this for consistency and correctness.

P 10 In 1-4: Please indicate these nine events or sub-phases by referring to table 2 in this sentence. The subsequent sentence explains the background of these events.

Decision letter (RSOS-192093.R0)

06-Apr-2020

Dear Dr Sahrom,

The editors assigned to your paper ("The use of yank-time signal as an alternative to identify kinematic events and define phases in human countermovement jumping") have now received comments from reviewers. We would like you to revise your paper in accordance with the referee and Associate Editor suggestions which can be found below (not including confidential reports to the Editor). Please note this decision does not guarantee eventual acceptance.

Please submit a copy of your revised paper before 29-Apr-2020. Please note that the revision deadline will expire at 00.00am on this date. If we do not hear from you within this time then it will be assumed that the paper has been withdrawn. In exceptional circumstances, extensions may be possible if agreed with the Editorial Office in advance. We do not allow multiple rounds of revision so we urge you to make every effort to fully address all of the comments at this stage. If deemed necessary by the Editors, your manuscript will be sent back to one or more of the original reviewers for assessment. If the original reviewers are not available, we may invite new reviewers.

- Data accessibility

It is a condition of publication that all supporting data are made available either as supplementary information or preferably in a suitable permanent repository. The data accessibility section should state where the article's supporting data can be accessed. This section should also include details, where possible of where to access other relevant research materials such as statistical tools, protocols, software etc can be accessed. If the data have been deposited in

an external repository this section should list the database, accession number and link to the DOI for all data from the article that have been made publicly available. Data sets that have been deposited in an external repository and have a DOI should also be appropriately cited in the manuscript and included in the reference list.

<http://datadryad.org/submit?journalID=RSOS&manu=RSOS-192093>

- **Competing interests**

- **Authors' contributions**

- **Acknowledgements**

- **Funding statement**

Kind regards,

Lianne Parkhouse

Editorial Coordinator

on behalf of Dr Derek Abbott (Associate Editor) and Pietro Cicuta (Subject Editor)

Reviewers' Comments to Author:

Reviewer: 1

Comments to the Author(s)

General Comment

The authors have conducted a thoughtful examination of a potential additive signal analysis

technique for a deeper understanding of vertebrae locomotion initially using the countermovement jump as the investigated movement. It was not apparent to this reviewer when reading the introduction that this was intended to be a technique used in conjunction with the current practice and I suggest that the authors reconsider the theoretical presentation to ensure the reader understands earlier that use of the Yank-time signal processing is to be used in cohort with other methods to improve the accuracy and reliability of event detection. This fact only becomes apparent in the conclusion. It is potentially outside of the scope for this manuscript but as the authors are seeking to instigate the use of new analytical method presenting the intra-individual reliability of the method for identifying the various events X-axis time variation and the Y-axis magnitude variation from the 3 maximal CMJ performed would aid the reader greatly.

It would be of interest to readers to know how this technique other than the ability to identify some events with greater certainty is better than applying statistical techniques such as spatial parametric mapping or functional principle component analysis to the GRF-time signal provides for a deeper understanding of the movement complexities? Granted the Yank-time signal as presented has strong alignment with sEMG onset and peak qualities (within analysis limits) and similarly with positional displacement data which also has it's own variation and limitations due to skin movement artefact etc, however all of these have been previously reported and accepted based on broad percentages of the movement time. Thus even SPA and fPCA all report back to a time normalisation process, so what is the use of Yank truly gaining the practitioner/scientist?

Specific Comment

Ln 23; suggest changing 'allowed' to 'informed' and 'greater' to 'deeper'

Pg 7 Ln 25; Please detail which model was used for the coordinate reconstruction?

Pg 8 Ln 12; Although at least 3 maximal efforts were performed only a CMJ single trial was analysed (selected from jump height), why not use an average of all jumps performed

Pg 11 Figure 2; As the authors are advocating for the use of the term Yank instead of RFD then it would be beneficial to the overall paper if this were consistently applied.

Pg 12 Table 2; Event 1) it is unclear how the new definition is any more accurate and it is arguably less objective than the traditional definition for the start of movement. Define with objective numerical terms what is required for a significant local maximum; Event 4) Considering that this event by the nature of the term used to describe it as Peak Rate of Force Development, it is a misnomer to indicate that it is not possible to integrate the signal for the resultant variable i.e. as has been shown for Event 2, which is then as described in the manuscripts introduction the commonality between Yank and Rate of Force Development (see also Journal of Experimental Biology 2019 222: jeb180414 doi: 10.1242/jeb.180414). Event 6a and 6b, would this better described as 6a: 0-50% of time from Event 5 to Event 7 (toe off) and 6b: 50-100% of time from Event 5 to Event 7 (toe off)?

Pg 14 Ln 4; the authors need to be more precise and provide objective information as to the tended alignment of the yank-time signal to the identifiable GRF-time signal (see general comment on reliability).

Pg 15 Ln 1; should this be referring to Table 2 as Table 4 is not providing movement onset definitions?

Pg 15 Ln 1-7; If there has been no change in the GRF and Yank is the integration of GRF with time, how is there a measurable change in the resultant Yank without it being the result of noise? Did the authors also apply any filtering techniques to the Yank as similarly applied to the other signals?

Pg 22 Ln 1-3; This statement is in contrast with that presented at Pg 8 Ln 12

Pg 22 Ln 3-6; Can the authors please detail the statistical approach they used to determine that there were no with-in subject variation to the shape of the GRF-time signal?

Pg 24 Ln 9; Grammar, should read, 'It is notable that...'

Pg 30 Table 6; check the grammar for the secondary criteria of the Unimodal-primary jump profile

Reviewer: 2
 Comments to the Author(s)

GENERAL COMMENTS

This paper examines the use of yank-time function as a way to identify kinematic events and define key phases and events in the countermovement jump.

The paper is very well written and the topic is important despite the prevalence of research on countermovement jumping across many years. The study identifies and addresses one of the key problems encountered by researchers in analysis force time data on countermovement jumping, namely, the difficulty in accurately defining key events on the time series such as the start of the countermovement since any error at this point carries forward and propagates errors in the constant of integration which will accumulate during integration. This is not an easy problem to resolve especially since neither the movement data or force data provide any correct value (gold standard) to test the accuracy of the proposed solution. The authors present a comprehensive analysis of how the yank-time function may offer a solution to the identification of the events and phases. The analysis includes kinematic and EMG data well as force time data to allow cross-validation of the effectiveness of the yank-time function. I commend the authors for a very thorough treatment of a challenging problem and I believe the content of the work will have good impact in the field. The manuscript is generally well written and well structured and therefore I only recommend various minor textual changes and clarifications (see specific comments below.

SPECIFIC COMMENTS

P3 ln 2: this is a verbatim repeat of the abstract... maybe some rewording would be appropriate

P 3 ln 9 : Perhaps omit however

P3 ln 25-26: passive expression... perhaps avoid by reversing the sentence clauses: "The vertical countermovement jump (CMJ) is a complex human movement task that has been well studied from a force-time perspective [6, 7]" I encourage the authors to consider similar instances of passive expression throughout the manuscript and reverse the clauses as this will improve the readability of the paper.

P4 Table 1: This table is very helpful in describing the derivative terminology however the reference suggests to me that this may not be an original source ... if this is not the original source can you please use the original source.

P8 ln 26: Teiger-Kaiser Energy Operator, the spelling is different in the cited reference. Please correct this for consistency and correctness.

P 10 ln 1-4: Please indicate these nine events or sub-phases by referring to table 2 in this sentence. The subsequent sentence explains the background of these events.

Author's Response to Decision Letter for (RSOS-192093.R0)

See Appendix A.

Decision letter (RSOS-192093.R1)

Dear Dr Sahrom,

It is a pleasure to accept your manuscript entitled "The use of yank-time signal as an alternative to identify kinematic events and define phases in human countermovement jumping" in its current form for publication in Royal Society Open Science.

Please ensure that you send to the editorial office an editable version of your accepted manuscript, and individual files for each figure and table included in your manuscript. You can send these in a zip folder if more convenient. Failure to provide these files may delay the processing of your proof.

on behalf of Dr Derek Abbott (Associate Editor) and Pietro Cicuta (Subject Editor)
openscience@royalsociety.org

Appendix A

The use of yank-time signal as an alternative to identify kinematic events and define phases in human countermovement jumping

Reviewer: 1

General Comment

The authors have conducted a thoughtful examination of a potential additive signal analysis technique for a deeper understanding of vertebrae locomotion initially using the countermovement jump as the investigated movement. It was not apparent to this reviewer when reading the introduction that this was intended to be a technique used in conjunction with the current practice and I suggest that the authors reconsider the theoretical presentation to ensure the reader understands earlier that use of the Yank-time signal processing is to be used in cohort with other methods to improve the accuracy and reliability of event detection. This fact only becomes apparent in the conclusion. It is potentially outside of the scope for this manuscript but as the authors are seeking to instigate the use of new analytical method presenting the intra-individual reliability of the method for identifying the various events X-axis time variation and the Y-axis magnitude variation from the 3 maximal CMJ performed would aid the reader greatly.

It would be of interest to readers to know how this technique other than the ability to identify some events with greater certainty is better than applying statistical techniques such as spatial parametric mapping or functional principle component analysis to the GRF-time signal provides for a deeper understanding of the movement complexities? Granted the Yank-time signal as presented has strong alignment with sEMG onset and peak qualities (within analysis limits) and similarly with positional displacement data which also has its own variation and limitations due to skin movement artefact etc, however all of these have been previously reported and accepted based on broad percentages of the movement time. Thus even SPA and fPCA all report back to a time normalisation process, so what is the use of Yank truly gaining the practitioner/scientist?

Reply to General Comment:

We appreciate the efforts of the reviewer, reviewing with great detail. The reviewer has raised several points which we hope we have successfully addressed. Given the above, the

first aim of the present study was to explore the use of yank, the time-derivative of force, to identify and describe important kinematic and kinetic events during a complex human movement (in our case, the CMJ) without the need for direct motion capture.

1a) Use of Yank-time signal with current methods.

The reviewer is correct, that the yank-time signal will need to be used in conjunction with additional and existing methods (integrating the force-time signal) for a comprehensive analysis as the existing methods alone may not be able to do so. However, as mentioned in our aims, one of the suggestions of this study is that the yank-time signal method proposed can be a suitable (1-step) alternative to current existing methods as it allows for the identification of certain events (e.g. Event 1) and other benefits (e.g. instantaneous RFD) that may be missed/overlooked using existing methods such as the force-time signal. Existing methods currently identify events directly from the force-time signal or by calculating several additional signals (e.g. displacement-time). We acknowledged that this aim on the use of yank-time signal as an alternative method might not be clear and have edited our Introduction to better clarify this, an example is below: -

“Given the above, the first aim of the present study was to explore the use of yank, the time-derivative of force, to identify and describe important kinematic and kinetic events during the CMJ without the need for direct motion capture as an alternative to existing methods.”

1b) Existing methods

As mentioned earlier, existing methods might fail to detect events during complex movements such as the CMJ. The reason existing methods might miss/overlook these events is that the events show as very small signal fluctuations that are difficult to identify by visual inspection or distinguish from noise. However, by calculating the time derivative, i.e. yank, these fluctuations are amplified into extrema (maxima and minima) which makes for easier event identification (these events can always be logically explained). For example, in Event 1, the rectified yank-time signal first increases (inflection point) due to joint flexion and then continues to increase as the joint continues to flex until it reaches a peak (maximum). This peak also reflects the first significant movement of the CoM since and is easily distinguished within the yank-time signal.

2) *“It would be of interest to readers to know how this technique other than the ability to identify some events with greater certainty is better than applying statistical techniques such as spatial parametric mapping or functional principle component analysis to the GRF-time signal provides for a deeper understanding of the movement complexities? & What is the use of Yank truly gaining the practitioner/scientist?”*

Understanding the unique movement patterns and strategies of a single animal/individual from force recordings is one of the main applications that we have in mind. Therefore, it is important to have a method that allows for this analysis of single individuals without the need to compare to cohort data (at least after a first study is completed in which the yank-time information is validated for use). Spatial parametric mapping and functional principal component analysis (PCA) largely allow descriptions of groups of individuals' movement patterns, however it is not a method that can be used for the analysis of single individuals, unlike the use of the yank-time signal, which we have shown allows for the identification of events that are likely to be missed/overlooked.

The ability to individually analyse the force-time signal of a specific individual is important for human locomotion because of the wide range of task adaptations that humans can adopt due to specific constraints or after exercise interventions (e.g. 100m sprinter versus an endurance runner) and therefore have very different movement patterns and strategies when performing the same movements (e.g. the CMJ) with the same objectives (e.g. maximising vertical displacement).

Having said that, we did consider the use of PCA techniques when comparing patterns between different groups of individuals (e.g. bimodal vs. unimodal), however we found that it was largely unnecessary and would have added further to the length and complexity of the final manuscript. We hope that it is clear that we were able to achieve our final aims without use of these techniques. We thank and appreciate the reviewer for suggesting the techniques such as spatial parametric mapping and it is something we are excited by and explore for future analyses especially, once we have the yank-time signals from a larger variety and sample of jumpers.

Reply to Specific Comment

Ln 23; suggest changing 'allowed' to 'informed' and 'greater' to 'deeper'

Thank you for the suggestion, we have made this change.

Pg 7 Ln 25; Please detail which model was used for the coordinate reconstruction?

Thank you for the suggestion, we have included the information.

Pg 8 Ln 12; Although at least 3 maximal efforts were performed only a CMJ single trial was analysed (selected from jump height), why not use an average of all jumps performed?

An important aim of the study was to explore the use of the yank-time signal, which is derived from a single ground reaction force-time signal. We therefore simply needed a method to consistently choose the jump to study from each individual, so we decided to take the best jump trial. We assumed that this jump would have been performed well by the individual and would therefore be reasonable to analyse.

We agree with the reviewer that an average of the best of several jumps would provide a better indication of an individual's "general" pattern of movement, and it will be interesting in future to compare outcomes when force-time traces are averaged and yank-time signal derived versus the averaging of yank-time signal variations for those jumps (i.e. whether to average before or after processing).

Pg 11 Figure 2; As the authors are advocating for the use of the term Yank instead of RFD then it would be beneficial to the overall paper if this were consistently applied.

We thank and agree with the reviewer on this suggestion. We have replaced the term RFD with yank as suggested or peak braking (for Event 4). We have only kept and introduced the RFD term much later in the manuscript at the point at which we highlight that yank-time signal is effectively the instantaneous RFD, which is of interest to some practitioners (e.g. sports scientists analysing athlete jump trials).

Pg 12 Table 2; Event 1) it is unclear how the new definition is any more accurate and it is arguably less objective than the traditional definition for the start of movement. Define with objective numerical terms what is required for a significant local maximum; Event 4) Considering that this event by the nature of the term used to describe it as Peak Rate of Force Development, it is a misnomer to indicate that it is not possible to integrate the signal for the resultant variable i.e. as has been shown for Event 2, which is then as described in the manuscripts introduction the commonality between Yank and Rate of Force Development (see also Journal of Experimental Biology 2019 222: jeb180414 doi: 10.1242/jeb.180414). Event 6a and 6b, would this better described as 6a: 0-50% of time from Event 5 to Event 7 (toe off) and 6b: 50-100% of time from Event 5 to Event 7 (toe off)?

Event 1:

We concur with the reviewer and have changed the definitions as such:

Traditional Definition	New Definition
When GRF_z decreases by an amount determined either as a 1) percentage, 2) standard deviation (e.g. 3 times of standard deviation), or 3) fixed amount (e.g. 10 N)	An inflection point (increase in yank) typically leading (continuously) to the first significant local maximum on the yank-time signal (Event 2).

Event 4: The reviewer is correct that Event 4 can be identified through integration but this is an additional step because the peak braking phase cannot be identified easily/directly from the force-time signal. We have also edited the definitions to better explain this.

Traditional Definition	New Definition
Not directly available from force-time signal. Peak braking phase can be identified from the peak -rate of force development (RFD) using several methods, 1) varying time interval or 3) instantaneous RFD (1 millisecond).	Most significant local maximum on the yank-time signal after Event 3. Instantaneous peak RFD.

Event 6a and 6b: This is a good suggestion as it does not require the reader to know or be able to identify the “upward phase”. We thank the reviewer for the suggestion and have made changes as such.

Traditional Definition	New Definition
N/A	0-50% of time from Event 5 to Event 7
N/A	51 to 100% of time from Event 5 to Event 7

Pg 14 Ln 4; the authors need to be more precise and provide objective information as to the tended alignment of the yank-time signal to the identifiable GRF-time signal (see general comment on reliability).

We concur that this sentence can be better phrased. We meant to clarify that in general we observed that the extrema (maxima and minima) within the yank-time signal aligned with significant events, which will be explained in more detail in the following sections of the paper. We have edited the phrase to:

“It is observed that the extrema (maxima and minima) within the yank-time signal tended to align closely with events identifiable on the GRF_z-time signal described in Table 2. These extrema, and their alignment with events, were subsequently explored in detail, as described in the following sections.”

Pg 15 Ln 1; should this be referring to Table 2 as Table 4 is not providing movement onset definitions?

Thank you for pointing this out. We have expanded and further clarified that it refers to the “Traditional Definitions” sections under Table 2.

“To assess the temporal correspondence of Event 1 identified using the yank-time signal relative to previously described methods (described under “Traditional Definition” in Table 2),”

Pg 15 Ln 1-7; If there has been no change in the GRF and Yank is the integration of GRF with time, how is there a measurable change in the resultant Yank without it being the result of noise? Did the authors also apply any filtering techniques to the Yank as similarly applied to the other signals?

Yes, the resultant yank-time signal was filtered using a Savitzky-Golay smoothing filter with a second-order polynomial, and we have now added the information in the paper.

“The yank-time data were subsequently rectified to allow for definitive identification of the crossover point, i.e., where yank was equal to zero, and then smoothed using a Savitzky-Golay smoothing filter with a second-order polynomial [28].”

Pg 22 Ln 1-3; This statement is in contrast with that presented at Pg 8 Ln 12

The best jump was selected for analysis. However, to allocate each individual to a group we considered their kinetic patterns of their best two jump trials, or the effective majority if they performed more than three jumps. However, we acknowledge that it can be written better and thank the reviewer for highlighting it. We have changed the sentence to:

“Based on the shape of the two best jump trials, or the effective majority if they performed more than three jumps, the subjects were categorised into one of three groups (Table 4).”

Pg 22 Ln 3-6; Can the authors please detail the statistical approach they used to determine that there were no with-in subject variation to the shape of the GRF-time signal?

We did not execute a statistical analysis for this purpose. The subjects were very well familiarised as they underwent a supervised familiarisation session followed by at least two unsupervised sessions before the testing day. Additionally, all subjects were physically active (at least twice a week) and experienced jumpers (deliberate exercise or sports sessions) and were consistent in their performances (observed during the familiarisation session). We examined the best two jumps and classified the jumper (as described in section 4.6). We found no significant differences/inconsistencies between the two best jumps. Therefore, we are confident of the classifications of the jumpers based on the shape. Also to share with the reviewers, when the subjects tried different movement patterns during the familiarisation sessions, for example, changing the countermovement

depth/speed, jumping what they perceive as faster or when they simply felt that the jump was not their “typical best jump”, the shape of the GRF-time signal did differ for some. These changes will be explored in a follow-up study where we also investigated the change in the shape of the force-time curve as they adopted different strategies. However, for now we are confident of the classifications of the jumpers based on the shape.

Pg 24 Ln 9; Grammar, should read, ‘It is notable that....’

Noted with thanks, we have made the relevant changes.

Pg 30 Table 6; check the grammar for the secondary criteria of the Unimodal-primary jump profile

Noted with thanks, we have changed to “There is a greater than 10% difference in normalised time between Event 5 and point of maximum GRF_z”

Reviewer: 2

General Comments

This paper examines the use of yank-time function as a way to identify kinematic events and define key phases and events in the countermovement jump.

The paper is very well written and the topic is important despite the prevalence of research on countermovement jumping across many years. The study identifies and addresses one of the key problems encountered by researchers in analysis force time data on countermovement jumping, namely, the difficulty in accurately defining key events on the time series such as the start of the countermovement since any error at this point carries forward and propagates errors in the constant of integration which will accumulate during integration. This is not an easy problem to resolve especially since neither the movement data or force data provide any correct value (gold standard) to test the accuracy of the proposed solution. The authors present a comprehensive analysis of how the yank-time function may offer a solution to the identification of the events and phases. The analysis includes kinematic and EMG data well as force time data to allow cross-validation of the

effectiveness of the yank-time function. I commend the authors for a very thorough treatment of a challenging problem and I believe the content of the work will have good impact in the field. The manuscript is generally well written and well-structured and therefore I only recommend various minor textual changes and clarifications (see specific comments below).

Reply to Specific Comments

P3 In 2: this is a verbatim repeat of the abstract... maybe some rewording would be appropriate

The reviewer is correct, and we have made some changes to wording:

“Detailed examinations of both the movement and muscle activation patterns used by animals and humans to complete complex tasks are difficult to obtain in many environments.”

P 3 In 9: Perhaps omit however

We thank the reviewer for the suggestion and have omitted “however”.

P3 In 25-26: passive expression... perhaps avoid by reversing the sentence clauses: "The vertical countermovement jump (CMJ) is a complex human movement task that has been well studied from a force-time perspective [6, 7]" I encourage the authors to consider similar instances of passive expression throughout the manuscript and reverse the clauses as this will improve the readability of the paper.

This is a valid point and we do see the benefits as the reviewer has suggested. We have made several changes in other sentences, but specifically for this sentence have decided to retain it since it continues on from the previous point “.... events during complex motion has yet to be determined”. Therefore, starting with a “A complex human movement tasks that has been well studied...” extends the previous sentence.

suggests to me that this may not be an original source ... if this is not the original source can yo zu please use the original source.

Thank you, you are correct. We have summarised this table from several sources and have now included the references.

“Kinematics’ describes the displacement or change in position. The higher-order time derivatives of displacement were summarised from the data from multiple sources [10–13]. Kinetic terms such as force and yank describe the rate of change in the momentum. *These terms have not yet been standardised [13].“

P8 In 26: Teiger-Kaiser Energy Operator, the spelling is different in the cited reference. Please correct this for consistency and correctness.

We thank the reviewers for pointing this out. We have corrected the spelling.

“For the purpose of EMG onset detection (EMG_{on}), EMG data were subjected to a Teager-Kaiser Energy Operator (TKEO), which highlights spikes in a signal by increasing sharpness while maintaining the signal’s temporal characteristics and has been shown to assist with onset detection with EMG signals [18–20].“

P 10 In 1-4: Please indicate these nine events or sub-phases by referring to table 2 in this sentence. The subsequent sentence explains the background of these events.

We thank the reviewer for pointing this out and have made the necessary changes. We have included the following sentence:

“These phases can be further sub-divided into nine distinct events or sub-phases (Table 2).“